# The First Case of *Granulicatella adiacens* Identified from a Resected Heart Valve by Next Generation Sequencing (NGS) in Poland

**DOI:** 10.3390/pathogens11030295

**Published:** 2022-02-25

**Authors:** Anna Podgórska, Maria Kordybach-Prokopiuk, Maria Jaworska-Wilczyńska, Piotr Hoffman, Katarzyna Biernacka, Krzysztof Kuśmierski, Robert Kuthan, Tomasz Hryniewiecki, Anna Lutyńska

**Affiliations:** 1Department of Medical Biology, National Institute of Cardiology, Stefan Cardinal Wyszyński State Research Institute, 42 Alpejska Street, 04-628 Warsaw, Poland; alutynska@ikard.pl; 2Department of Congenital Heart Diseases, National Institute of Cardiology, Stefan Cardinal Wyszyński State Research Institute, 42 Alpejska Street, 04-628 Warsaw, Poland; mkordybach@ikard.pl (M.K.-P.); phoffman@ikard.pl (P.H.); kbiernacka@ikard.pl (K.B.); 3Department of Valvular Heart Disease, National Institute of Cardiology, Stefan Cardinal Wyszyński State Research Institute, 42 Alpejska Street, 04-628 Warsaw, Poland; mwilczynska@ikard.pl (M.J.-W.); thryniewiecki@ikard.pl (T.H.); 4Department of Cardiac Surgery and Transplantology, National Institute of Cardiology, Stefan Cardinal Wyszyński State Research Institute, 42 Alpejska Street, 04-628 Warsaw, Poland; kkusmierski@ikard.pl; 5Department of Medical Microbiology, Medical University of Warsaw, 5 Chałubińskiego Street, 02-004 Warsaw, Poland; rkuthan@wum.edu.pl

**Keywords:** infective endocarditis, *Granulicatela adiacens*, nutritionally variant streptococci (NVS), next-generation sequencing (NGS), aortic valve

## Abstract

In this report, we describe the course and successful treatment of a case of complicated infective endocarditis (IE). A patient presented with a high-grade, irregular fever with chills lasting at least 2 months along with dyspnoea, chest pain, fatigue, weight loss, and night sweats during the previous 3 months. As well as cardiac congenital disorders, he was found to have *Granulicatella adiacens* infective aortic valve endocarditis, presumably transmitted from the oral cavity niche. Validated metagenomic 16S rDNA next generation sequencing was used to perform taxonomic identification, allowing for specific adequate antibiotic therapy instead of empiric therapy. This paper highlights the critical role of rapid taxonomic identification of nutritionally variant streptococci and the benefit of proper IE treatment in avoiding relapses or fatal complications.

## 1. Introduction

Nutritionally variant streptococci (NVS) growth in standard blood culture depends on pyridoxal or cysteine supplementation. Culture-negative infective endocarditis (IE) cases sustain the main clinical problems since NVS are slow-growing, unusual, and require additional media in order to be identified. In 1961, Frenkel and Hirsch described NVS as fastidious organisms that display satellite growth around colonies of other bacteria [1]. Compared to NVS, streptococci, enterococci, and staphylococci are easily grown, well recognized, and described as IE aetiological agents [2,3]. Based on 16S rRNA gene sequencing, Collins and Lawson [4] proposed that NVS are members of the *Granulicatella* and *Abiotrophia* genera, which have rarely demonstrated a specific microbiological cause of IE. As a life-threatening disease, IE is a prominent cause of morbidity and mortality, and cardiac valvular dysfunctions are known to be important risk factors in contracting IE. Dental procedures are also frequently associated with the transmission of mouth commensal bacteria, such as NVS, from the oral cavity to the blood vessels [5]. NVS are part of the healthy oral flora, intestinal tract, and urogenital system. *Granulicatella adiacens* are observed frequently in the oral cavity and in dental plaque, endodontic infections, and dental abscesses. IE induced by NVS is associated with high morbidity rates, frequent treatment failures, high incidence of relapses, and is a source of growing antibiotic resistance [6]. Novel sequencing techniques may improve the identification of IE etiology when conventional tests fail to yield valid results. To the best of our knowledge, no other case of IE caused by the *G. adiacens* case has been reported in Poland.

## 2. Material and Methods

### 2.1. Clinical Patient Status

A 20-year-old patient who had been diagnosed with a bicuspid aortic valve in childhood was admitted to the Department of Congenital Heart Diseases of the Cardinal Stefan Wyszyński Institute of Cardiology in Warsaw in May 2020 with an implanted mechanical aortic valve and was suspected of having infective endocarditis (IE). The patient suffered from a high-grade, irregular fever with chills lasting at least 2 months along with dyspnoea, chest pain, fatigue, weight loss, and night sweats during the previous 3 months. An echocardiographic examination performed in 2019 in another hospital revealed severe aortic regurgitation with an enlarged left ventricle (LVEDD: left ventricular end-diastolic diameter, 61 mm). After transthoracic echocardiography (TTE), transesophageal echocardiography (TEE) and computed tomography (CT) were performed and the patient was referred for aortic valve replacement.

### 2.2. Laboratory Examinations

The patient was examined for respiratory and cardiac parameters coupled with biochemical and culture microbiology testing.

### 2.3. Valve Culture

Each dissected valve sample was dissolved in 100 mL BHI (brain–heart infusion) medium and incubated at 37 °C with 5% CO_2_ for a maximum of 10 days. Aliquots of 100 µL of BHI cultured valve sample were removed and spread on the surface of solid microbiological media (bioMérieux, Marcy l’Etoile, France): Columbia agar with 5% defibrinated sheep blood, MacConkey agar, bile esculin agar, mannitol salt agar, Schaedler agar with 5% defibrinated sheep blood, and supplemented chocolate agar with PolyViteX^TM^. For the fungal culture, Sabouraud agar supplemented with gentamicin and chloramphenicol was used. All bacterial media were incubated at 37 °C under aerobic conditions except for the Schaedler medium and supplemented chocolate agar with PolyViteX^TM^, which were incubated under anaerobic and microaerophilic conditions, respectively. Aerobic, anaerobic, and microaerophilic cultures were checked for growth after 24 and 48 h. The Sabouraud medium was incubated at 30 °C under aerobic conditions for 7 days and checked every 24 h for any growth.

### 2.4. Blood Culture

Three sets of culture bottles, with each set including a bottle for aerobic microorganisms (BD BACTEC™ Plus Aerobic medium) and a bottle for anaerobic microorganisms (BD BACTEC™ Lytic Anaerobic medium), were used. Each bottle was inoculated with 10 mL of blood drawn from the patient at their bedside. Each set of blood samples were collected at 30–60 min intervals. Samples were collected before antibiotic treatment or administration of the next antibiotic. All bottles were incubated in the BACTEC FX (Becton Dickinson, Heidelberg, Germany) blood culture system for 7 days after inoculation. If no bacterial growth was detected within 7 days, the blood culture was considered negative.

### 2.5. DNA Extraction

The resected valve was collected from the patient with diagnosed IE (according to the modified Duke’s criteria) in cooperation with the Department of Cardiac Surgery and Transplantology in the National Institute of Cardiology in Poland. Resected valve sections were subjected to culturing in the Department of Microbiology and Hospital Infections Monitoring in the National Institute of Cardiology in Poland.

The resected specimen was cut into two equal-sized pieces using sterile scissors in a biosafety cabinet. One piece of tissue was randomly selected for immediate culturing (as described above), while the other piece of tissue was snap frozen at −80 °C for metagenomic sequencing and Sanger validation.

The frozen valve was thawed at room temperature for 30 min and cut into the smallest possible pieces with sterile scissors. Bacterial DNA was extracted from approximately 25 mg of tissue using the Genomic Mini AX Bacteria Kit mod. 7 (A&A Biotechnology, Gdańsk, Poland), which was modified by applying additional enzyme digesting steps with lysozyme, lysostaphin, and mutanolysin. The purity and concentration of the total bacterial DNA were determined through 260/280 nm absorbance measures using the NanoDropOne spectrophotometer (Thermo Fisher Scientific, Waltham, MA, USA). Quantification of the DNA was also performed using 1% agarose gel electrophoresis at a potential of 100 V (Consort EV265 Range Power Supply, Alpha Metrix Biotech, Rödermark, Germany). After separation, the DNA isolate was visualized under UV light after staining with an appropriate dye (Infinity VX2, Vilber Lourmat, Collégien, France).

Enrichment of DNA isolated from the valve section was performed using the NEB Next Microbiome DNA Enrichment Kit (New England BioLabs, Frankfurt, Germany). DNA was also extracted from the water used to ensure the absence of contaminating DNA. In addition, DNA was extracted from the valve tissue of patients without an IE diagnosis.

### 2.6. 16S rDNA Next-Generation Sequencing

Paired-end sequencing libraries were constructed according to the Illumina protocol for preparing libraries. The first step involved the amplification of a DNA sample using specific primers to amplify a region of interest, which was followed by attaching specific overhanging adapters. The gene-specific sequences used in this protocol targeted the 16S V3 and V4 region [7]. The full-length primer sequences used in the protocol to target this region were as follows: 16S Amplicon PCR Forward Primer: 5′-TCGTCGGCAGCGTCAGATGTGTATAAGAGACAGCCTACGGGNGGCWGCAG-3′; 16S Amplicon PCR Reverse Primer: 5′-GTCTCGTGGGCTCGGAGATGTGTATAAGAGAC AGGACTACHVGGGTATCTAATCC-3′.

The second step involved the application of MagSi-NGS PREP Plus beads (Illumina, San Diego, CA, USA) to purify the 16S V3 and V4 amplicons from free primers and primer dimers. The next step involved attaching dual indexes and Illumina sequencing adapters using the Nextera XT Index Kit (Illumina, San Diego, CA, USA). After the second clean-up step, the final library was quantified, normalized, and pooled. Finally, multiplexed paired-end sequencing (2 × 300 bp reads) of the 16S rDNA amplicons was performed using Illumina MiSeq technology (MiSeq Reagent Kit v3 600 cycles, Illumina, San Diego, CA, USA).

High-quality sequencing data were obtained by removing low-quality reads (species with identified reads ≤10), short reads, and duplicated reads as well as adapter contaminations. Human sequence data were mapped to a human reference genome (GRCh38.p11) using Burrows-Wheeler Alignment (BWA) software. All human sequences were removed, and the remaining sequencing data were aligned to the NCBI nucleotide database using the Basic Local Alignment Tool (BLAST) [https://blast.ncbi.nlm.nih.gov, accessed on 14 November 2017].

### 2.7. Sanger Sequencing on Ribosomal DNA Extracted from the Heart Valve

Confirmation of NGS-positive findings was performed with capillary Sanger sequencing using the previously described primers targeting the 550-bp sections of the 16S V3 and V4 regions. Simultaneously, the primer pair was used to amplify a 422-bp fragment unique to *G. adiacens* (sense: 5′-GGTTTATCCTTAGAAAGGAGGT-3′; antisense:

5′-GAGCATTCGGTTGGGCACTCTAG-3′) [8]. Each 20 μL of the PCR reaction mixture contained 20 ng of total or microbial-enriched DNA as a template. The PCR reaction was carried out using the following protocols: an initial denaturation step was performed at 94 °C for 10 min, followed by 31 cycles of denaturation (94 °C, 30 s), annealing (53 °C, 1 min), extension (72 °C, 1 min), and a final elongation of 7 min at 72 °C.

The PCR product then underwent agarose gel electrophoresis and was visualized under UV light. Sanger sequencing was performed on a 3130xL Genetic Analyzer (Applied Biosystems, Waltham, MA, USA) for validation. Finally, the sequences obtained from Sanger sequencing were aligned to the NT database with NCBI BLAST online software.

### 2.8. Ethics

The study protocol was approved by the hospital ethics committee (study number IK-NPIA-0021-56/1643/17 at 14 November 2017). The patient provided written informed consent to participate in the study.

## 3. Results

During auscultation, the patient showed a systolic–diastolic murmur. No other relevant findings were elicited. Blood tests were as follows: procalcitonin 0.08 ng/mL (ref. value 0–0.5 ng/mL), CRP 1.4 mg/dL (ref. value 0–0.5 mg/dL), WBC 39100/µl, haemoglobin 11.8 g/dL, NT-proBNP 11,276 pg/mL, ALT 184 U/l, AST 104 U/l, LDH 317U/l. Echocardiograms (TTE and TEE) revealed multiple vegetations and a perforation of the anterior mitral leaflet with severe mitral regurgitation. While it was difficult to evaluate the movement of the mechanical valve discs, severe paravalvular leaks with multiple vegetation were identified in the posterior part of the aortic ring. The left ventricle was enlarged (LVEDD: 70 mm) with preserved systolic function (LVEF: left ventricle ejection fraction, 60%). The right ventricle was also enlarged (RVIT: right ventricular inflow tract, 53 mm) with decreased systolic function (TAPSE: tricuspid annular plane systolic excursion, 12 mm). Based on these findings, a diagnosis of infective endocarditis was made. Empiric antibiotic therapy with vancomycin, gentamicin, and rifampicin was initiated. The necessity of performing cardiac surgery was confirmed by a multidisciplinary team, and the mechanical mitral valve and aortic homograft were implanted surgically. The postoperative care was complicated by multi-organ failure; as a result, arterial–venous ECMO was applied. A cavity after an empty abscess in the mitral aortic curtain was intraoperatively removed. The postoperative echocardiography examination showed normal function of the mechanical aortic valve, and a mild leakage in the posterior section of the aortic ring and moderate mitral regurgitation was found. The laboratory test indicated microcytic anemia (hemoglobin 8.6 g/dL); therefore, a gastroscopy and a colonoscopy were performed. After no signs of hemorrhage were found in the digestive tract, ferrum supplementation was prescribed. Antibiotic therapy was continued, adjusting doses appropriate to kidney function and determining the blood concentration. Parameters of inflammation as well as parameters of kidney and liver function improved slowly and the patient’s condition improved. In a control echocardiography examination, the systolic function of the borderline volume of the left ventricle (LVEDV: left ventricular end-diastolic volume, 150 mL) had decreased (LVEF, 30–35%). Two filiform echoes attached to the mitral subvalvular apparatus were noted. An abdominal USG showed splenomegaly without focal changes.

Experimental antibiotic therapy was applied on 26 May; three blood samples were collected at half-hour intervals, and resected aortic and mitral valve samples were determined to be negative during 7 days of culturing. The patient received the following treatment: gentamicin i.v. (3 × 0.08 g for 32 days and 1 × 0.08 g for the following 25 days), vancomycin i.v. (2 × 1 g for 32 days followed by a four-day break, then 1 × 0.5 g/2 days during the following 21 days), and rifampicin (2 × 0.6 g for 29 days followed by a four-day break, then 2 × 0.6 g during the following 25 days). Blood cultures collected at the end of six weeks of antibiotic therapy were still negative. However, during the four-day breaks from vancomycin and rifampicin treatment as well as 6 months following hospitalization, the patient complained again of high-grade, irregular fever (39.5 °C) and fatigue. To confirm the reliability of the microbiological tests, a NGS metagenomic study was implemented. Bacterial DNA was isolated from heart valve sections, a source of the entire bacteria, and underwent an NGS metagenomic procedure that was confirmed by Sanger sequencing and showed *G. adiacens* as a causative bacterium. The sequence was matched with *G. adiacens* along with 24 partial sequences of 16S RNA available in the BLAST tool library and exhibited 99% identity with *G. adiacens* isolate MT459366.1. Finally, ceftriaxone with gentamicin was applied for treating IE induced by *G. adiacens*, which resulted in a bacteriological cure and was continued for 4 weeks.

## 4. Discussion

Infective endocarditis (IE) due to nutritionally variant streptococci (NVS) is a very rare condition and is often associated with negative blood cultures. IE caused by NVS has a higher rate of complications compared to endocarditis caused by other streptococci [9,10]. Over 100 cases of NVS infective endocarditis have been reported in the literature so far. NVS microorganisms have been found to be responsible for up to 6% of all cases of streptococcal endocarditis [11]. IE induced by NVS is characterized by frequent relapses and a higher morbidity compared with infections induced by other species of streptococci. A clinical decision for rapid surgical intervention may be needed, especially in penicillin-resistant cases [12]. Nevertheless, our case was culture-negative, and the surgical decision was made based on TTE and TEE results that clearly showed the presence of vegetations. Until metagenomic sequencing results became available, empirical therapy was initiated. After the pathogen was identified, a change in antibiotic treatment was appropriate, since previous data have showed that *G. adiacens*-infected IE patients experience a high rate of treatment success using ceftriaxone and gentamicin [11,13]. After changing to *G. adiacens*-specific targeted therapy, the patient became afebrile and was in a good condition, symptomatically. No relapses were found on subsequent follow-up visits during the following six months.

The portal of initial entry of this taxonomic group of bacteria into the endovascular system can be related to dental procedures. These organisms are assumed to have originated from the oral cavity. Since the patient displayed poor oral hygiene and had undergone dental procedures several times before the clinical symptoms of IE developed, we can assume that he became infected before hospital admission. The 16S rDNA NGS analysis successfully revealed fastidious and unusual *G. adiacens* in the resected heart valve of the patient. About 85% of identified bacterial genomes contain more than 1 copy of 16S rDNA; therefore, the method is able to identify bacteria genomes with high sensitivity [14]. It is possible to obtain a complete metagenomics analysis of a clinical sample in less than 30 h using Illumina sequencing platforms [15]. The application of NGS is of increasing clinical interest because this approach allows for faster pathogen identification compared to standard microbiological techniques. In our paper, the case described circumstances such as blood cultures and an excised valve before antibiotic treatment and they remained negative on day 7. The 16S rRNA gene is ubiquitous to all bacterial domains, allowing for the construction of ever-expanding sequence databases, which are critical to the definite identification of bacteria. The value of metagenomic sequencing should not be ignored, particularly in cases of culture-negative IE. Genetic analysis of microbial DNA from heart valve tissue has been included recently into the British Society for Antimicrobial Chemotherapy BSAC and ESC IE guidelines [16]. Even the current version of the most authoritative and widely used Bergey’s Manual of Systematic Bacteriology utilizes 16S RNA gene sequences to characterize and identify many kinds of bacteria [17]. Clinicians should be very careful with reports of culture-negative results and should consider that culture-based identification is time consuming and frequently fails to produce relevant data (since the media spectra available are restricted in many cases) within a critical time period for making rapid therapeutic decisions. Since classical symptoms of IE induced by rare and fastidious microorganisms are not typical, metagenomics studies are the only option for clinicians to provide patients with the proper treatment necessary to overcome the fatal complications. It should be emphasized that the time required to diagnose IE using metagenomic analysis may create a fast switch from empirical therapy to a specific one. Since a wide spectrum of molecular tools are currently available, especially for clinically significant cases such as IE, clinicians, clinical microbiologists, and molecular biologists should increase their level of cooperation. Close communication between clinicians and the medical laboratory is necessary to connect their findings with clinical pictures.

## 5. Conclusions

Blood and valve culture-negative endocarditis is a life-threatening disease that is extremely difficult to diagnose and treat. Molecular biology techniques have a higher sensitivity than traditional microbiological methods for detecting IE, especially when IE is related to fastidious or difficult-to-culture microorganisms (e.g., HACEK bacteria and defective streptococci—*Gemella*, *Granulicatella* and *Abiotrophia* sp.—*Propionibacterium acnes*, *Candida* sp.) [18]. Our report clearly showed that there was a clinical indication to perform nucleotide amplification techniques and NGS in situations where classical microbiology methods had failed. We concluded that routine high confidence identification of a microbial community, at least on the genus level, could be obtained for a large number of medically important organisms, especially those undescribed or fastidious, by utilizing a 16S rRNA gene database generated at a particular medical health center. According to the WHO, despite innovations in diagnostic tools and systems for identifying microbial infections, fast, specific, sensitive, and accurate molecular diagnostics in humans have not yet been routinely adopted. IE caused by NVS presents tremendous diagnostic and therapeutic challenges even in the modern medical era.

## Data Availability

Not applicable.

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
