# Peer review of "The First Case of Granulicatella adiacens Identified from a Resected Heart Valve by Next Generation Sequencing (NGS) in Poland"

_pathogens, 2022, doi:10.3390/pathogens11030295_

Round 1

Reviewer 1 Report

  • The English needs some improvement, see my proposals.
  • There are some statements in the manuscript concerning the appropriate antibiotic therapy for an infective endocarditis (IE) caused by Granulicatella adiacens which need an addition and correction:
    1. Introduction, lines 42-43: To the sentence ending “... is a source of growing antibiotic resistance.” add the following relevant reference should be added:

Prasidthrathsint, K.; Fisher, M.A. Antimicrobial Susceptibility Patterns among a Large, Nationwide Cohort of Abiotrophia and Granulicatella Clinical Isolates. J. Clin. Microbiol. 2017, 55, 1025-1031, doi:10.1128/jcm.02054-16.

  1. The citation of reference no. 9 (Lin et al., 2007) in this context (discussion, lines 209-211) is not correct, since this paper does not support the statement in the authors’ text. In contrast, as can be found in the review of Prasidthrathsint et al. (see above) a higher level of penicillin resistance in adiacens is often associated with ceftriaxone resistance. In the case of the patient described in the manuscript, the treatment including ceftriaxone worked, but this cannot be concluded to be a rule from the literature cited by the authors!
  • In the following some proposals for corrections are given:

General: Type names of microorganisms always in italic, e. g. Granulicatella adiacens!

      The time course of the patient’s stay in the hospital is unclear: In the Materials and Methods section (line 50), the authors state that the patient was admitted to the hospital in July 2020. However, in the Results section (line 179) they report that “Experimental antibiotic therapy was applied on 26th May”.

Abstract: Line 1, change to “... successful treatment of a case of complicated ...”

Line 20: omit “crucial”

Introduction:    Line 34-35, change to “... specific microbiological cause ...”

Materials and Methods: Line 57: Omit “results”.

Line 62: omit “fluid”; line 67 change to “... and supplemented chocolate agar PolyVitexTM (biomérieux).” Line 70: change accordingly!

Line 85: change to “... from the patient ...

Lines 106-106: change to “... from the water used to ensure the absence of contaminating DNA.” Line 117: correct the spaces from the primer sequence!

Lines 129-130: change to “... were aligned to the NCBI nucleotide database using the Basic Local Alignment Tool (BLAST)” Please add the appropriate reference for BLAST here or add the link to the NCBI nucleotide BLAST website if you used this!

Line 132: change to “... positive finding was performed with capillary Sanger sequencing using ...”. Line 144: correct to “... NCBI BLAST online software.”

Results section: Line 161, correct to “... gentacicin, ...”

Line 173, change to “... liver function improved slowly ...”

Line 182, correct to “gentamicin”

Discussion: Line 207: Abbreviation “TE” not explained in the preceding text!

Line 221: change to “... identify bacterial genomes ...”

Line 228: change as follows “... sequence databases, which are critical to the definite identification ...”

Lines 241-242: change to “... analysis may create a fast swift from empirical therapy to a specific one.”

Lines 242-246: I propose to change these sentences as follows: “... clinicians and clinical microbiologists should increase their level of cooperation. Close communication between clinicians and the medical laboratory is necessary to connect their findings with clinical pictures.”

Conclusions: Line 251, change to “... of difficult-to-culture microorganisms ...”.

Line 253: change to “... to perform nucleotide amplification techniques and NGS ...”.

Line 260: omit “frequently and”

Reviewer 2 Report

In my opinion it is a well written, clinicaly important case report of a practical value.

I just would like to ask if

  • There were (except poor dental condition) any other risk factors of IE in patient's past medical history?
  • In lines 194-195 authors write: Finally, ceftriaxone with gentamycin was applied for treating IE induced by G. adiacens, resulting in a bacteriological cure. Finally means when- can a specific time be given?  It would be of value to say how long treatment was continued.
  • I suggest to give ref. values for procalcitonin and CRP - they can differ between labs
